# Effect of Vegetable Juice, Puree, and Pomace on Chemical and Technological Quality of Fresh Pasta

**DOI:** 10.3390/foods10081931

**Published:** 2021-08-20

**Authors:** Jinghong Wang, Margaret Anne Brennan, Charles Stephen Brennan, Luca Serventi

**Affiliations:** 1Department of Wine, Food and Molecular Biosciences, Lincoln University, P.O. Box 85084, Lincoln 7647, New Zealand; Jinghong.wang@lincolnuni.ac.nz (J.W.); Margaret.Brennan@lincoln.ac.nz (M.A.B.); charles.brennan@rmit.edu.au (C.S.B.); 2Riddet Institute, Palmerston North 4442, New Zealand; 3School of Science, RMIT University, P.O. Box 2474, Melbourne, VIC 3001, Australia

**Keywords:** quality, texture, physicochemical, vegetable pasta, colour

## Abstract

Vegetable pasta is a premium product, and its consumption may deliver health benefits by increasing vegetable intake. This study investigated the replacement of semolina with juice, puree, and pomace of spinach and red cabbage. The effect of replacement on chemical composition, cooking performance (cooking loss, swelling index, water absorption), texture quality (elasticity, firmness), and colour was evaluated. The cooking loss of pasta made with spinach juice and spinach puree at 1 g/100 g substitution was the same as the control, while all other samples had a higher cooking loss. Spinach pasta had a higher breaking force but lower breaking distance in the tensile test than the control, while red cabbage pasta had a lower breaking force and breaking distance. Spinach juice fortified pasta was firmer than the control. Red cabbage juice pasta was less firm than other forms of fortified pasta at 1 g/100 g substitution level. Spinach and red cabbage juice are better colorants than puree or pomace as they change the colour of the pasta more dramatically at the same substitution level. Cooking performance and texture quality of spinach juice pasta were better than other samples, which indicates a premium pasta product for the food industry.

## 1. Introduction

Pasta is a staple cereal food worldwide and it is a good vehicle for delivering functional ingredients [1,2]. Vegetables contain many health-promoting phytochemicals that traditional pasta lacks [3]. Those phytochemicals include dietary fibre, vitamins, polyphenols, carotenoids, glucosinolates, and minerals. Even though consumers are aware of the health benefits of consuming vegetables, their ingrained eating habits prevent them from a sufficient vegetable intake [4]. Hence, incorporating vegetables in staple foods such as pasta or bread may be a good option.

Vegetable pasta has been studied by many researchers [5,6]. An inferior cooking and sensory quality of vegetable pasta have been frequently reported compared to the traditional product [7,8,9,10]. Quality changes typically include increased cooking loss (CL), decreased firmness, and elasticity [1]. Authors substituted semolina for vegetable powder [7,11,12]. The powder is not only involved in nutrition loss due to air oven-drying [13], but also related to solid particle-size associated quality impact [14]. Hence, pasta enriched with other forms of vegetables, such as puree, juice, and pomace, were investigated in this study.

There are a few studies that have used other forms of vegetables to enhance pasta. For example, Gull et al. [15] added carrot pomace powder to a pasta formula, while Simonato et al. [16] used 5–10% olive pomace to fortify pasta. Carini et al. [17] added carrot juice to pasta and found that carrot juice pasta had similar extensibility and CL compared to the control, while carrot flour enriched pasta had a very high CL (more than 8 g/100 g) and lower extensibility, which indicates inferior quality. This study’s limitation is that the actual substitution level (based on the dry matter) of carrot flour is much higher than carrot juice so that it is not a like for like comparison. Rakhesh et al. [18] made use of carrot, spinach, tomato, and beetroot puree to fortify pasta and found a decreased CL and improved texture of the resultant pasta. This study lacks comparison with the powder form and the description of puree-semolina mixing procedure is unclear. Juice and puree addition also have limitations when combined with pasta. Juice contains a very low level of solids, mostly around 5–15% for fresh vegetable juice [19]. Thus, achieving a high substitution level based on dry matter may be impossible for juice and puree due to excessive hydration, which can cause large lump formation, resulting in difficulties in successful extrusion [20,21]. The water content of juice and puree makes them more difficult to store and transport and may cause an increased cost for the food industry.

This project investigated the optimum method of vegetable fortification to produce vegetable-enriched pasta with better texture and cooking quality. The aim was to compare the key chemical composition, cooking performance, and texture quality of vegetable fortified pasta produced using vegetable juice, puree, and pomace. Our preliminary study showed that vegetable powder shows no significant difference with vegetable puree when added to pasta in key technical tests such as elasticity, firmness, and cooking loss. However, the nutritional quality (e.g., antioxidant ability) of powder enriched pasta was lower than puree enriched pasta. This may be due to the oven drying used to produce the vegetable powder. Therefore, the powder was altered to puree in our study. Two kinds of leafy vegetables, spinach (*Spinacia oleracea* L.) and red cabbage (Brassica oleracea convar. capitata var. capitata f. rubra), were selected in this study. Spinach is cheap and widely available. It is considered to have antioxidant and antidiabetic effects [22]. Spinach is also widely accepted by the food industry to produce commercially available green pasta. Red cabbage is nutritious as it is high in fibre and antioxidant phytochemicals [23]. Red cabbage materials in this study created a novel purple coloured pasta.

## 2. Materials and Methods

### 2.1. Raw Material

Semolina (Sun Valley Foods Ltd., Hamilton, New Zealand, labeled protein = 10.7 g/100 g, Fibre = 2.1 g/100 g, sodium = 10 mg/100 g), fresh spinach, and red cabbage were brought from the local market (New World Supermarket, Lincoln, New Zealand).

### 2.2. Vegetable Preparation

Spinach and red cabbage were washed thoroughly, their roots were removed with a sharp knife, the stem and leaf were put into the juicer (Model: Oscar Neo DA 1000; NATURE’S WONDERLAND Ltd., Brisbane, Australia), and the pomace and juice were collected separately. The vegetable juice was placed into a separate glass jar with a cap and stored at −18 °C until use. The pomace was spread in a backed tray and put into an oven to dry at 60 °C for 7 h. The dried pomace was then ground to a powder with a coffee grinder for 10 s twice, and the resultant pomace powder was stored in a Ziplock plastic bag at room temperature. The spinach puree and red cabbage puree were produced by mixing juice and fresh pomace together using a blender (Nutri-bullet NBO7200-1210DG; Capitalbrands Ltd., Boston, MA, USA), then the spinach puree and red cabbage were collected in a glass jar with a cap and stored at −18 °C. Before use, the puree was defrosted and put into the blender again to homogenise.

### 2.3. Pasta Preparation

Pasta was prepared using a lab-scale pasta machine (Model: MPF15N235M; Firmer., Ravenna, Italy) with 20 holes of 2.25 mm diameter. The vegetable pomace fortified pasta was prepared by mixing pomace with semolina in a pasta machine, and then 40 °C water was added according to the manual of the device to extrude the pasta. The formula is shown in Table 1. The puree and juice were defrosted and warmed to 40 °C in a water bath. Then, the puree, or juice, and 40 °C water were added to semolina in the pasta machine to extrude the pasta according to Table 1. The substitution level of juice and puree pasta is based on dry matter, according to the solid content measurement of the raw material. 1% substitution of juice, and 2% substitution of puree is the substitution level that can be both achieved by spinach and red cabbage material.

### 2.4. Proximate Analysis

Solid and moisture content was measured using the oven-dry method (105 °C), according to AACC [24]. The protein content of raw material and pasta was determined by Dumas total N methods (Elemental analyser Vario MAX CN, Frankfurt, Germany), and a conversion factor of 6.25 was applied to both pasta and raw materials to convert N to protein %. It should be noted that the conversion factor for different vegetables might vary as total N included non-amnio acid N like nitrate and N from nucleic acids [25]. Thus, the protein results are proximate, especially for the raw material. Total starch was measured by AOAC official method 966.11 using Megazyme total starch kit. Ash content of raw materials and pasta samples was measured according to AACC [24].

### 2.5. Cooking Performance

#### 2.5.1. Optimal Cooking Time

Optimal cooking time (OCT) was measured according to AACC [24]. A total of 20 units of 5 cm pasta strands were put into 300 mL boiling water. The OCT was evaluated by taking a strand every 30 s and squeezing it between two transparent glass slides until the white core disappeared.

#### 2.5.2. Pasta Cooking Procedure

Aliquots of 10 g of pasta were cooked in 600 mL of boiling water at OCT, then rinsed with 100 mL of cold water and stained for 30 s to measure cooking loss (CL), swelling index (SI), and water absorption index (WAI).

#### 2.5.3. Cooking Loss

The CL was measured according to AACC [24]. Cooking water was collected by a stainless-steel vessel and dried in an air oven at 105 °C until a constant weight was reached. The residue was weighed and reported as gram residue per 100 g raw material.

#### 2.5.4. Swelling Index and Water Absorption Index

SI and WAI were evaluated according to Desai et al. [26] with slight modification. 10 g of pasta was cooked to OCT and weighed after wash and stain, recorded as P_c_. Then, the cooked pasta was dried at 105 °C until a constant weight was reached, recorded as P_cd_. SI and WAI can be calculated with the following Equations (1) and (2):SI = (P_c_ − P_cd_)/P_cd_(1)
WAI = (P_c_ − P_u_)/P_u_ ∗ 100(2)
P_u_ is the weight of uncooked pasta, P_c_ is the weight of cooked pasta, P_cd_ is the weight of dried, cooked pasta.

### 2.6. Texture Measurement

The firmness, breaking distance, and breaking force was measured by a Texture Analyser (TA.XT2; Stable Micro systems, Godalming, UK) with a 5 kg load cell. The pasta was cooked to OCT as described above before test. Firmness test is according to Approved Method 66-50 [24], with some modifications. Five strands of cooked pasta were placed on a flat metal plate. A noodle blade was used to compress the cooked pasta strands. The test parameters were set as test speed = 0.2 mm/s, post test speed = 10 mm/s, and distance of 5 mm. Tension test setting was according to [27]. The A/SPR spaghetti/noodle rig (Settings: Pre-test speed, 3 mm/s; test speed, 3 mm/s; initial distance, 10 mm; Final Distance 120 mm) was used in the test. Data are represented as the mean of nine measurements from triplicate cooking batches.

### 2.7. Colour Measurement

A portable colour meter (Minolta Chroma Meter CR210; Minolta Camera Co., Japan) was used to measure cooked (to OCT) and uncooked pasta. Each pasta was measured nine times from the triplicate cooking batches, and the result was expressed as L * (brightness range from 100 to 0), a * (redness–greenness range from 128 to −128), b * (yellowness–blueness range from 128 to −128). The instrument was calibrated using a standard white tile (L * = 98.03, a * = −0.23, b * = 0.25).

### 2.8. Statistical Analysis 

All experiments were performed in triplicate except for what has been mentioned. All data were statistical analysed by one-way ANOVA, and the difference was evaluated by the Duncan test. SPSS (version 16) was used to perform data and figures.

## 3. Results and Discussion

### 3.1. Proximate Composition of Vegetable Pasta

The protein, moisture, and ash content of spinach and red cabbage pasta are shown in Table 2. The protein content in pasta is essential as it is the key of pasta structure. In pasta, gluten protein can be described as the backbone, with starch granules trapped in it playing a crucial function in pasta structure [28]. It is widely accepted that this structure is mainly maintained by disulfide bonds with the help of other non-covalent interactions like hydrogen bonds and ionic bonds [29,30,31]. It is suggested that protein-rich material may result in protein–protein interaction and form a more cohesive structure, thus helping the gluten form a homogeneous pasta structure [32]. Table 2 (a) shows that spinach raw material is rich in protein, ranked by protein content the spinach pasta is as follows SJ > SPU > SPO based on the dry weight. As a result, all the uncooked spinach pasta shows a significantly higher (*p* < 0.05) protein content than the control. Lisiewska et al. [33] reported that raw spinach contains 36 ± 12 mg/100 g cysteine content, around 1.5% of its total amino acid composition. Cysteine can provide sulfhydryl groups to form disulphide bonds during dough formation [34]. It indicates that protein from spinach may positively impact the formation of a gluten network and pasta quality. For cooked pasta, higher protein content was observed compared to raw. A similar trend was found by Manthey and Hall III [35], who use buckwheat bran flour to enrich pasta. It is possibly due to the leaching of starch into the cooking water, increasing the proportion of protein content in the pasta. Although for SJ1, the uncooked pasta shows significantly higher protein content than the control, the cooked one shows no difference (*p* < 0.05). This may be because the juice sample contains more soluble protein that may be lost during cooking. The cooked red cabbage pasta has a significantly lower protein content than control except for RCPU2, possibly because red cabbage raw material has a lower protein content (RCPO contains 11.06 g/100 g compared to SPO of 23.91 g/100 g).

The total starch content of vegetable enriched pasta decreased with increased vegetable substitution. Cooking did not show any significant difference in total starch composition (*p* < 0.05) of vegetable enriched pasta. The ash content of foods is mainly inorganic metal compounds [36]. Perssini, Sensidoni, Pollini, and De Cindio [30] found that sodium chloride content increases the strength and solid-like semolina-flour dough behaviour via optimization of ionic strength. McCann and Day [37] found that salt delays the formation of the gluten network by reducing the rate of gluten hydration. Tang et al. [38] found that salt content can increase the strength of the disulfide bond in flour Raman gluten dough as less free SH groups are detected. Thus, the ash content may influence pasta quality. Table 2 shows spinach raw material characteristics with higher ash content in every form compared with red cabbage raw material. The addition of vegetable material increased the ash content of vegetable pasta significantly (*p* < 0.05) in every sample. This result is similar to Prabhasankar et al. [39], who used Japanese seaweed to fortify pasta. The ash content indicates a higher mineral content in those samples. Cooking causes a decreased ash content of vegetable pasta. It may be because some metal in ash is present in a water-soluble form and is lost during cooking. Desai, Brennan, and Brennan [26] found similar trends showing that cooked fish powder fortified pasta had a lower ash content than before cooking.

### 3.2. Cooking Quality of Vegetable Pasta

Optimal cooking time (OCT), cooking loss (CL), swelling index (SI), and water absorption index (WAI) are crucial cooking quality attributes of pasta [20]. Those attributes are strongly influenced by the protein–starch matrix formed during cold extrusion [31]. A good quality pasta has a compact protein–starch matrix, which slows the diffusion of the water to the starch core and inhibits amylose leaching into cooking water, giving a longer OCT and a decreased CL [40]. Table 3 shows that red cabbage content increases the cooking loss significantly. The CL of red cabbage pasta ranged from 4.767 to 6.163 g/100 g, compared to 4.399 g/100 g of the control sample. The CL values of the spinach juice pasta and spinach puree pasta shows no significant difference at 1 g/100 g substitution level versus the control. Other spinach pasta samples show increased cooking loss (from 4.447 to 5.920 g/100 g) compared to control. The increased CL indicates a weaker gluten matrix, which may be caused by fibre disruption, competition for water between gluten protein and other compounds (such as water-soluble fibre and soluble salt), and a dilution of gluten, which is caused by the substitution of semolina with vegetable material. The pasta with red cabbage pomace had a higher cooking loss than puree or juice sample at the same substitution level (RCPO1 > RCPU1, RCPO1 > RCJ1, RCPO2 > RCPU2). This may be because it contained less protein compared to pasta made with puree or juice, as shown in Table 2. The protein content and their properties can influence the gluten network formation and pasta structure [20]. The higher protein content in juice and puree can potentially interact with gluten, hence decreasing the disruptive effect caused by fibre and diluted gluten. Carini, Curti, Spotti, and Vittadini [17] found that carrot juice pasta has a much lower CL than pasta with carrot flour. However, the substitution level of carrot juice and carrot flour in that research was not standardised. Kowalczewski et al. [41] report the CL of fresh potato juice fortified pasta is lower than that fortified by spray-dried potato juice. All the vegetable pasta in this study had a CL lower than 8 g/100 g, which is a widely agreed maximum value for consumer acceptability [20,42].

The OCT is not changed in the vegetable pasta of all 1 g/100 g, 2 g/100 g samples, possibly because at low substitution levels the gluten network is not significantly changed to create a measurable impact. However, at a substitution level of 10 g/100 g, SPO10 and RCPO10 have a shorter optimal cooking time (Table 3). The decreased OCT may be caused by decreased water absorption (from 81.27 g/100 g of control to 74.11 g/100 g of SPO10 and 73:80 g/100 g of RCPO10). Similar results were found by Aravind et al. [43] using inulin (soluble fibre) to enrich pasta, and a lower OCT was reported. Cárdenas-Hernández et al. [44] also found OCT was decreased when amaranth flour and amaranth leaves and carboxymethylcellulose were added to semolina to produce pasta. In contrast, Foschia et al. [45] found an increased OCT when using 15 g/100 g dietary fibre (such as long-chain inulin, psyllium, or Glucagel) to substitute semolina.

The swelling index (SI) and water absorption index (WAI) reflect the amount of water absorbed at OCT. Table 3 shows that all spinach pasta samples show the same swelling index compared to control. Similar results were reported by Yadav et al. [46], which shows spinach pasta has no significant difference in water absorption versus control. Red cabbage pomace samples (RCPO1, RCPO10) show a lower WAI (*p* < 0.05) compared to control, while red cabbage juice and puree pasta show no significant SI and WAI difference versus control. It is potentially because the components of the red cabbage pomace have less affinity for water than the components of the red cabbage juice or puree. The results of WAI and SI of RCPO pasta are consistent with Sun-Waterhouse, Jin, and Waterhouse [6], who found that elderberry juice pasta absorbs less water than other samples. In contrast, water absorption increase was observed in turnip pasta, tomato pasta, and carrot pasta [46], as well as broad bean flour fortified pasta [47]. This study may indicate that the SI and WAI of vegetable pasta is dependent on the intactness or strength of the gluten network and the water-binding capacity of vegetable components.

### 3.3. Texture and Colour of Vegetable Pasta

Pasta texture plays an essential role in overall quality and consumer acceptance [15,26]. Elasticity is an important texture profile that is considered to be conferred by gliadins that interact non-covalently with high modular weight glutenin subunits [31]. Elasticity (breaking distance and breaking force) of spinach pasta and red cabbage pasta is shown in Figure 1a,b, respectively. Spinach pasta has a higher breaking force (*p* > 0.05, except SPO1 and SPO2 insignificantly higher) than control. Meanwhile, red cabbage addition shows no significant influence on the breaking force of RCJ1, RCPU1, RCPU2, and RCPO1 and decreased breaking force was observed for RCPO2 and RCPO10. SJ1 and SPU1 have the same breaking distance as control while other spinach pasta and all red cabbage pasta were characterized by lower breaking distance. Juice fortified pasta shows a higher breaking distance compared to puree and pomace fortified pasta (SJ1 > SPO1, SPU2, SPO2, SPO10 significantly, RCJ1 > RCPU1, RCPO1, RCPO2, RCPU2 & RCPO10). At a higher substitution level of 10 g/100 g. The breaking distance of SPO10 and RCPO10 decreased dramatically. The decreased breaking distance indicates a weakened structure. Lu et al. [48] found a lower breaking force compared to control when adding white button mushroom powder (5–15%) and porcini mushroom powder (10–15%) to pasta, respectively. The same authors reported no significant change in breaking force when incorporating 5–15% shiitake mushroom powder to durum wheat to produce pasta. Foschia, Peressini, Sensidoni, Brennan, and Brennan [27] found that breaking force was decreased when durum wheat was substituted with 15 g/100 g dietary fibre (inulin, psyllium and oat material).

Firmness is a measure of the force needed to compress pasta strands between teeth, and is an indicator of protein matrix integrity after cooking, which is dependent on the quality of gluten fraction [20]. Figure 1c,d show the firmness of spinach pasta and red cabbage pasta, respectively. The spinach pasta has a greater firmness than the control (except SPU1, SPO1, and SPO2). At the same substitution level, SJ1 has a greater firmness than SPU1 and SPO1. One possible reason for this is that SJ1 has fewer solid components. Those components may form discontinuities or cracks inside the pasta and result in a weakened structure. Red cabbage pasta firmness was equal to or lower than the control, while spinach pasta firmness was equal to or higher than the control. This is possibly because spinach pasta has a higher protein content than red cabbage pasta (as shown in Table 2). The higher protein content may contribute to a stronger protein structure, thus mitigating the disruptive effect of dietary fibre on the gluten network. This assumption is consistent with Petitot, Boyer, Minier, and Micard [8], who substituted 35% of semolina with split pea or faba bean and reported a significantly firmer pasta with a higher protein content. Jayawardena, Morton, Brennan, and Bekhit [32] used 10–25% protein-rich beef lung powder added to durum wheat, and the resultant pasta had a significantly higher firmness and breaking force. The firmness of RCJ1 is the lowest of all tested samples, possibly because of more water swelling (see Table 3 swelling index) by the starch granules, which in turn created a softer texture. Foschia, Peressini, Sensidoni, Brennan, and Brennan [27] found that incorporating short-chain inulin leads to a dramatic decrease in pasta firmness and increased water absorption. Gull, Prasad, and Kumar [15] reported a significantly lower firmness than control when 2–10% carrot pomace was added to the pasta formula.

It may be assumed that the texture profile of vegetable pasta is dependent on the vegetable components. Some components such as fibre and sugar may adversely affect the overall texture and cooking quality as they influence the water absorption, thus causing a change in the hydration process of the starch granules and the gluten network. Fibre particles dilute the gluten and therefore also contribute to gluten network disruption and potentially weaken the structure. Other components, such as protein, may generally have some beneficial effects such as strengthening the gluten network and other interactions to enhance the structure, such as increasing the firmness and breaking force. The overall texture and cooking quality are dependent on the balance of such adverse and beneficial effects from vegetable components. SJ1 in this study provides outstanding cooking and texture quality, with the identical cooking loss, water absorption, and breaking distance compared with control. It also has higher firmness and breaking force than control, thus produces al dente products with a firm, elastic texture. A low substitution level (1 g/100 g according to dry matter), juice form (lower solid particles), and higher protein content than durum wheat may contribute to its distinctive texture quality.

Colour results of vegetable pasta are shown in Table 4. The colour of vegetable pasta is strongly influenced by vegetable addition. Red cabbage pasta has lower brightness and yellowness (less L * and b * value) and higher redness (increased a * value) compared to the control. When comparing the different forms of vegetables, the juice’s dye effect is stronger than puree or pomace as RCJ1 (both raw and cooked) has lower brightness and yellowness and more redness than RCPU1 and RCPO1. After cooking, the red cabbage pasta tends to be brighter but less red and yellow. Possibly due to the fact that the phytochemicals that provide the colour are water-soluble and leach into the cooking water. Chigurupati et al. [49] found that red cabbage colour is water-soluble and sensitive to pH change. It was found that the red cabbage colour changed from purple to deep blue when pH changed from acid to neutral. This could explain why cooked red cabbage pasta tends to be bluer (lower b * value except for RCPU1) and the cooking water presents a slightly blue colour as the water boils, leading to acid evaporation [50]; thus, the pH of cooking water tends to be neutral. Spinach addition leads to the decreased brightness, redness, and yellowness (L *, a *, b *, respectively) of the resultant pasta. Cooking procedure decreases the lightness, greenness, and yellowness of spinach pasta. Interestingly, the yellowness decrease of cooked spinach pasta is much lower than control (from 29.58 to 13.74 of control vs. 13.05 to 11.03 of SJ1, for example), indicating that spinach reduces the yellowness decrease during cooking. This is consistent with Nisha et al. [51], who found that thermal treatment causes a decrease in lightness and greenness but improved yellowness of spinach puree.

## 4. Conclusions

The results show that the juice, puree, and pomace of vegetables behave differently when incorporated into a pasta formulation. Those differences are plausible due to heterogeneous compositions in the varied forms of vegetables. At a low substitution level (1−2 g/100 g), juice, puree, and pomace can all be used to produce pasta with acceptable cooking performance and texture quality. Juice fortified pasta has lower cooking losses and better elasticity compared to puree and pomace fortified pastas. Among all pasta samples in this study, the cooking performance and texture quality of spinach juice pasta were better than other vegetable pastas and comparable or even better than control. This is probably due to its higher protein (cysteine-rich) composition and low substitution level (less gluten dilution and structure interruption). The study may indicate that vegetable juice with high protein content, such as spinach juice, can be used to produce premium pasta products for the food industry.

## Figures and Tables

**Figure 1 foods-10-01931-f001:**
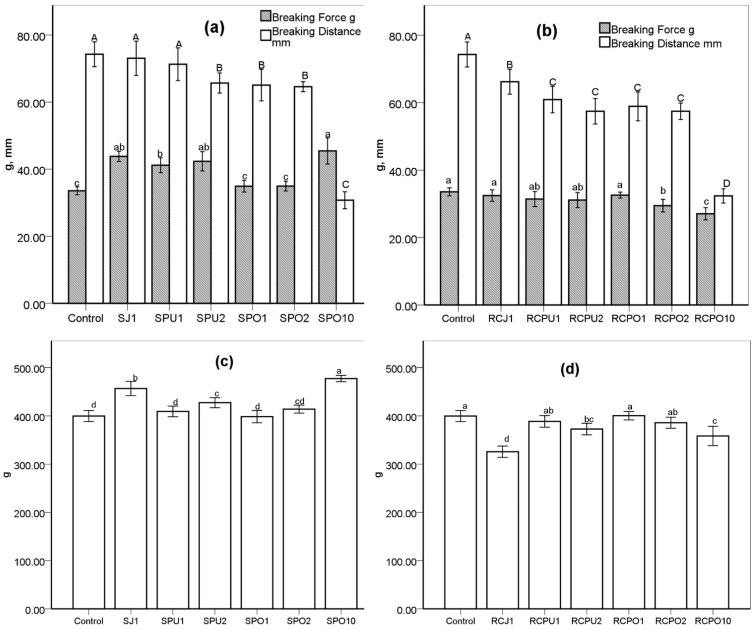
(**a**) Elasticity of spinach pasta, (**b**) Elasticity of red cabbage pasta, (**c**) Firmness of spinach pasta, (**d**) Firmness of red cabbage pasta. SJ, SPU, SPO represent spinach juice, spinach puree, spinach pomace, respectively; RCJ, RCPU, RCPO represent red cabbage juice, red cabbage puree, red cabbage pomace, respectively; 1, 2, and 10 is the substitution level (g/100 g) based on the dry weight. C: control sample. Error bars present the standard deviation of replicates. The same letter mean values are not significantly different from each other (*p* > 0.05).

**Table 1 foods-10-01931-t001:** Pasta formula to produce every 130 g pasta.

Pasta Type	Semolina g	Water g	Vegetable Amount g	Water from Vegetable g	Dry Matter from Vegetable g	Substitution Level %
C	100	30	0	0	0	0
SJ1	99	9.51	21.49	20.49	1	1
SPU1	99	11.63	19.37	18.37	1	1
SPU2	98	8.73	23.27	21.27	2	2
SPO1	99	30	1	0	1 *	1
SPO2	98	30	2	0	2 *	2
SPO10	90	30	10	0	10 *	10
RCJ1	99	6.93	24.07	23.07	1	1
RCPU1	99	19.43	11.57	10.57	1	1
RCPU2	98	8.85	23.15	21.15	2	2
RCPO1	99	30	1	0	1 *	1
RCPO2	98	30	2	0	2 *	2
RCPO10	90	30	10	0	10 *	10

SJ, SPU, and SPO represent spinach juice pasta, spinach puree pasta, and spinach pomace pasta, respectively; RCJ, RCPU, and RCPO represent red cabbage juice, red cabbage puree, and red cabbage pomace, respectively; 1, 2, and 10 is the substitution level (g/100 g) based on the dry weight. C: control sample. * the water content of pomace (see Table 2) is neglected in this study because it is less than 14%, which is close to that of semolina.

**Table 2 foods-10-01931-t002:** Proximate chemical analysis of vegetable pasta.

(a) Spinach pasta and spinach raw material								
	**Protein g/100 g dry matter**		**Total Starch g/100 g dry matter**		**Moisture g/100 g Material**		**Ash g/100 g dry matter**	
	**Uncooked**	**Cooked**	**Uncooked**	**Cooked**	**Uncooked**	**Cooked**	**Uncooked**	**Cooked**
Raw material								
Semolina	12.58 ± 0.11	N/A	71.42 ± 0.51	N/A	10.95 ± 0.10	N/A	0.95 ± 0.04	N/A
SJ	38.56 ± 0.06	N/A	N/A	N/A	95.35 ± 0.04	N/A	21.82 ± 0.02	N/A
SPU	31.17 ± 0.07	N/A	N/A	N/A	91.41 ± 0.07	N/A	17.40 ± 0.05	N/A
SPO	23.91 ± 0.13	N/A	N/A	N/A	12.95 ± 0.02	N/A	13.29 ± 0.01	N/A
Spinach pasta								
C	12.49 ± 0.01 ^d,B^	12.96 ± 0.01 ^de,A^	70.35 ± 0.39 ^a,A^	70.24 ± 0.26 ^a,A^	36.16 ± 0.70 ^ab^	64.58 ± 0.23 ^a^	0.68 ± 0.01 ^g,A^	0.44 ± 0.01 ^f,B^
SJ1	12.76 ± 0.01 ^c,B^	13.00 ± 0.06 ^cd,A^	69.60 ± 0.62 ^b,A^	69.46 ± 0.50 ^b,A^	35.71 ± 0.06 ^b^	64.40 ± 0.24 ^a^	0.89 ± 0.00 ^d,A^	0.55 ± 0.01 ^d,B^
SPU1	12.75 ± 0.01 ^c,B^	13.04 ± 0.01 ^c,A^	69.72 ± 0.60 ^b,A^	69.48 ± 0.46 ^b,A^	35.36 ±0. 25 ^b^	64.47 ± 0.15 ^a^	0.84 ± 0.00 ^e,A^	0.55 ± 0.00 ^d,B^
SPU2	12.96 ± 0.01 ^b,B^	13.24 ± 0.06 ^b,A^	68.06 ± 0.41 ^c,A^	68.41 ± 0.45 ^c,A^	35.67 ± 0.22 ^b^	64.50 ± 0.26 ^a^	1.11 ± 0.01 ^b,A^	0.65 ± 0.01 ^b,B^
SPO1	12.78 ± 0.01 ^c,B^	12.89 ± 0.01 ^e,A^	69.44 ± 0.48 ^b,A^	69.29 ± 0.37 ^b,A^	36.71 ± 0.68 ^ab^	65.00 ± 0.16 ^a^	0.81 ± 0.00 ^f,A^	0.58 ± 0.00 ^c,B^
SPO2	12.77 ± 0.01 ^c,B^	13.06 ± 0.01 ^c,A^	68.13 ± 0.62 ^c,A^	68.57 ± 0.52 ^c,A^	35.62 ± 0.25 ^b^	64.61 ± 0.05 ^a^	0.93 ± 0.01 ^c,A^	0.48 ± 0.00 ^e,B^
SPO10	14.13 ± 0.03 ^a,B^	14.44 ± 0.01 ^a,A^	61.39 ± 0.43 ^d,A^	60.93 ± 0.36 ^d,A^	37.27 ± 0.64 ^a^	64.53 ± 0.90 ^a^	1.94 ± 0.01 ^a,A^	1.14 ± 0.01 ^a,B^
(b) Red cabbage pasta and red cabbage raw material								
Raw material								
RCJ	19.23 ± 0.01	N/A	N/A	N/A	95.85 ± 0.00	N/A	10.18 ± 0.01	N/A
RCPU	16.23 ± 0.08	N/A	N/A	N/A	91.36 ± 0.02	N/A	8.26 ± 0.02	N/A
RCPO	11.06 ± 0.01	N/A	N/A	N/A	13.14 ± 0.07	N/A	5.50 ± 0.04	N/A
Red cabbage pasta								
C	12.49 ± 0.01 ^b,B^	12.96 ± 0.01 ^a,A^	70.35 ± 0.39 ^a,A^	70.24 ± 0.26 ^a,A^	36.16 ± 0.70 ^c^	64.58 ± 0.23 ^b^	0.68 ± 0.01 ^f,A^	0.44 ± 0.01 ^e,B^
RCJ1	12.46 ± 0.03 ^b,B^	12.82 ± 0.04 ^b,A^	68.97 ± 0.56 ^b,A^	68.40 ± 0.51 ^b,A^	36.41 ± 0.41 ^bc^	64.68 ± 0.53 ^ab^	0.78 ± 0.01 ^c,A^	0.46 ± 0.00 ^d,B^
RCPU1	12.41 ± 0.01 ^c,B^	12.63 ± 0.03 ^d,A^	68.88 ± 0.21 ^b,A^	68.46 ± 0.42 ^b,A^	37.45 ± 0.13 ^abc^	64.79 ± 0.37 ^b^	0.76 ± 0.00 ^d,A^	0.50 ± 0.01 ^c,B^
RCPU2	12.56 ± 0.01 ^a,B^	12.91 ± 0.01 ^a,A^	67.50 ± 0.58 ^c,A^	67.32 ± 0.57 ^c,A^	36.63 ± 0.75 ^abc^	64.60 ± 0.72 ^b^	0.84 ± 0.02 ^b,A^	0.54 ± 0.02 ^b,B^
RCPO1	12.30 ± 0.01 ^d,B^	12.59 ± 0.01 ^d,A^	68.74 ± 0.47 ^b,A^	68.41 ± 0.42 ^b,A^	37.77 ± 0.33 ^ab^	65.83 ± 0.04 ^a^	0.73 ± 0.01 ^e,A^	0.50 ± 0.01 ^c,B^
RCPO2	12.56 ± 0.05 ^a,B^	12.72 ± 0.01 ^c,A^	67.45 ± 0.57 ^c,A^	67.30 ± 0.40 ^c,A^	37.30 ± 0.36 ^abc^	65.42 ± 0.46 ^ab^	0.78 ± 0.01 ^c,A^	0.55 ± 0.00 ^b,B^
RCPO10	12.41 ± 0.01 ^c,B^	12.77 ± 0.03 ^bc,A^	60.34 ± 0.58 ^d,A^	59.96 ± 0.50 ^d,A^	37.94 ± 0.45 ^a^	64.30 ± 0.14 ^b^	1.16 ± 0.01 ^a,A^	0.95 ± 0.01 ^a,B^

SJ, SPU, and SPO represent spinach juice, spinach puree, and spinach pomace, respectively; RCJ, RCPU, and RCPO represent red cabbage juice, red cabbage puree, and red cabbage pomace, respectively; N/A means not tested. 1, 2, and 10 is the substitution level (g/100 g) based on the dry weight. C: control sample. Results expressed as Mean ± standard deviation calculated from triplicate measurements. Protein starch and ash results are based on a dry weight basis. Values within a column in the same sub-table followed by the same superscripted letters are not significantly different from each other (*p* > 0.05), values followed by the same superscripted capital letter are not significantly different between cooked and uncooked samples according to the ANOVA-Duncan test.

**Table 3 foods-10-01931-t003:** Cooking performance of vegetable pasta.

	Optimal Cooking Time (Mins: Second)	Cooking Loss (g/100 g)	Swelling Index (g Water/g Dry Pasta)	Water Absorption Index (g/100 g)
Spinach Pasta				
C	7:00	4.399 ± 0.063 ^de^	1.863 ± 0.065 ^a^	81.27 ± 1.42 ^a^
SJ1	7:00	4.367 ± 0.065 ^e^	1.801 ± 0.019 ^a^	80.62 ± 1.17 ^a^
SPU1	7:00	4.447 ± 0.092 ^de^	1.814 ± 0.012 ^a^	81.92 ± 1.22 ^a^
SPU2	7:00	4.800 ± 0.026 ^c^	1.817 ± 0.015 ^a^	82.09 ± 0.28 ^a^
SPO1	7:00	4.503 ± 0.015 ^d^	1.858 ± 0.013 ^a^	80.86 ± 2.18 ^a^
SPO2	7:00	5.001 ± 0.062 ^b^	1.826 ± 0.004 ^a^	81.93 ± 0.90 ^a^
SPO10	6:30	5.920 ± 0.781 ^a^	1.821 ± 0.073 ^a^	74.11 ± 2.67 ^b^
Red Cabbage Pasta				
C	7:00	4.399 ± 0.063 ^e^	1.863 ± 0.065 ^ab^	81.27 ± 1.42 ^ab^
RCJ1	7:00	4.767 ± 0.021 ^d^	1.927 ± 0.004 ^a^	82.16 ± 0.78 ^a^
RCPU1	7:00	4.803 ± 0.070 ^d^	1.832 ± 0.043 ^b^	80.07 ± 1.59 ^abc^
RCPU2	7:00	4.943 ± 0.068 ^c^	1.878 ± 0.025 ^ab^	81.29 ± 0.34 ^ab^
RCPO1	7:00	5.083 ± 0.379 ^b^	1.840 ± 0.030 ^b^	77.67 ± 1.54 ^c^
RCPO2	7:00	5.067 ± 0.076 ^b^	1.826 ± 0.057 ^b^	79.03 ± 1.85 ^bc^
RCPO10	6:15	6.163 ± 0.067 ^a^	1.801 ± 0.012 ^b^	73.80 ± 0.96 ^d^

SJ, SPU, and SPO represent spinach juice, spinach puree, and spinach pomace, respectively; RCJ, RCPU, and RCPO represent red cabbage juice, red cabbage puree, and red cabbage pomace, respectively; 1, 2, and 10 is the substitution level (g/100 g) based on the dry weight. C: control sample. Results expressed as Mean ± standard deviation calculated from triplicate measurements. Values within a column of the same kind of pasta followed by the same superscripted letter are not significantly different from each other (*p* > 0.05) according to the ANOVA-Duncan test.

**Table 4 foods-10-01931-t004:** Colour characteristics of cooked and uncooked pasta enriched with spinach and red cabbage.

	Uncooked				Cooked			
	L	a	b	Colour Example	L	a	b	Colour Example
Spinach Pasta								
C	65.38 ± 0.40 ^a^	−0.36 ± 0.07 ^a^	29.58 ± 0.18 ^a^		61.68 ± 0.30 ^a^	−0.66 ± 0.02 ^a^	13.74 ± 0.02 ^a^	
SJ1	43.04 ± 0.21 ^f^	−9.75 ± 0.09 ^g^	13.05 ± 0.18 ^c^		40.16 ± 0.19 ^e^	−7.71 ± 0.15 ^g^	11.03 ± 0.06 ^b^	
SPU1	49.56 ± 0.11 ^c^	−9.59 ± 0.02 ^f^	14.79 ± 0.03 ^b^		43.63 ± 0.04 ^c^	−6.85 ± 0.07 ^f^	10.39 ± 0.26 ^c^	
SPU2	45.66 ± 0.09 ^e^	−7.30 ± 0.03 ^e^	10.68 ± 0.03 ^e^		40.06 ± 0.41 ^e^	−6.25 ± 0.13 ^e^	9.27 ± 0.16 ^d^	
SPO1	51.02 ± 0.46 ^b^	−5.12 ± 0.20 ^b^	12.69 ± 0.11 ^d^		46.49 ± 0.31 ^b^	−4.79 ± 0.19 ^d^	7.93 ± 0.38 ^e^	
SPO2	48.54 ± 0.44 ^d^	−7.07 ± 0.09 ^d^	10.23 ± 0.29 ^f^		41.43 ± 0.26 ^d^	−4.54 ± 0.27 ^c^	7.36 ± 0.23 ^f^	
SPO10	38.74 ± 0.12 ^g^	−5.78 ± 0.06 ^c^	7.60 ± 0.17 ^g^		29.83 ± 0.08 ^f^	−3.45 ± 0.09 ^b^	4.95 ± 0.04 ^g^	
Red Cabbage Pasta								
C	65.38 ± 0.40 ^a^	−0.36 ± 0.07 ^g^	29.58 ± 0.18 ^a^		61.68 ± 0.30 ^a^	−0.66 ± 0.02 ^e^	13.74 ± 0.02 ^a^	
RCJ1	41.93 ± 0.74 ^f^	8.85 ± 0.05 ^a^	−7.17 ± 0.10 ^g^		46.46 ± 0.58 ^g^	3.12 ± 0.12 ^c^	−8.61 ± 0.16 ^e^	
RCPU1	47.54 ± 0.39 ^c^	5.91 ± 0.02 ^d^	−2.22 ± 0.02 ^e^		54.26 ± 0.33 ^c^	−1.26 ± 0.03 ^f^	−1.38 ± 0.38 ^b^	
RCPU2	44.55 ± 0.35 ^e^	6.30 ± 0.02 ^c^	−5.49 ± 0.02 ^f^		50.07 ± 0.85 ^e^	4.47 ± 0.18 ^b^	−8.68 ± 0.28 ^e^	
RCPO1	49.59 ± 0.38 ^b^	3.25 ± 0.01 ^f^	3.59 ± 0.04 ^b^		58.02 ± 0.09 ^b^	−1.69 ± 0.07 ^g^	−2.31 ± 0.40 ^c^	
RCPO2	45.29 ± 0.32 ^d^	4.01 ± 0.01 ^e^	2.05 ± 0.02 ^c^		53.23 ± 0.11 ^d^	2.05 ± 0.04 ^d^	−5.31 ± 0.27 ^d^	
RCPO10	36.55 ± 0.12 ^g^	6.82 ± 0.03 ^b^	−1.59 ± 0.50 ^d^		47.12 ± 0.06 ^f^	5.75 ± 0.06 ^a^	−5.04 ± 0.04 ^d^	

SJ, SPU, and SPO represent spinach juice pasta, spinach puree pasta, and spinach pomace pasta, respectively; RCJ, RCPU, and RCPO represent red cabbage juice, red cabbage puree, and red cabbage pomace, respectively; 1, 2, and 10 is the substitution level (g/100 g) based on the dry weight. C: control sample. Results expressed as Mean ± standard deviation calculated from ten measurements. L a b colour is converted to R G B colour through https://www.nixsensor.com/free-color-converter/ (accessed on 22 July 2021) and colour was output through EXCEL. While the colour convertor can only input integer colour number, the generated example colour is proximate. Values within a column from the same kind of pasta followed by the same superscripted letter are not significantly different from each other (*p* > 0.05), according to the ANOVA- Duncan test.

## Data Availability

The data presented in this study are available in the Table 2, Table 3 and Table 4 and Figure 1 within the article.

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
