# Peer review of "Effect of Vegetable Juice, Puree, and Pomace on Chemical and Technological Quality of Fresh Pasta"

_foods, 2021, doi:10.3390/foods10081931_

Round 1

Reviewer 1 Report

The present article faces the partial substitution of semolina with vegetable products in pasta formulations. The argument is very well presented, and the research is well developed and carried out in a rigorous manner. There are just a few points I wanted to discuss with the authors.

1) In the manuscript there are no information about semolina under study characteristics (protein amount, gluten quantity and quality). If it is possible at least to report the data shown in the product label this could be helpful, since protein amount is a very important information.

2) How did you choose the amount of water in every recipe? This is a very important parameter so an explanation of how you dosed it is required.

3) Regarding the references I think authors should consider also some other more recent studies about semolina dough ingredient characterization and substitution to compare with their formulation. Examples are:

Ahmad, N., Ur-Rehman, S., Shabbir, M. A., Shehzad, M. A., & Roberts, T. H. (2018). Fortification of durum wheat semolina with detoxified matri (Lathyrus sativus) flour to improve the nutritional properties of pasta. Journal of food science and technology55(6), 2114-2121.

De Pasquale, I., Verni, M., Verardo, V., Gómez-Caravaca, A. M., & Rizzello, C. G. (2021). Nutritional and functional advantages of the use of fermented black chickpea flour for semolina-pasta fortification. Foods, 10(1), 182.

Fanari, F., Desogus, F., Scano, E. A., Carboni, G., & Grosso, M. (2020). The effect of the relative amount of ingredients on the rheological properties of semolina doughs. Sustainability, 12(7), 2705.

Romano, A., Ferranti, P., Gallo, V., & Masi, P. (2021). New ingredients and alternatives to durum wheat semolina for a high quality dried pasta. Current Opinion in Food Science.

Schettino, R., Pontonio, E., & Rizzello, C. G. (2019). Use of fermented hemp, chickpea and milling by-products to improve the nutritional value of semolina pasta. Foods, 8(12), 604.

Xie, L., Nishijima, N., Oda, Y., Handa, A., Majumder, K., Xu, C., & Zhang, Y. (2020). Utilization of egg white solids to improve the texture and cooking quality of cooked and frozen pasta. LWT, 122, 109031.

4) Attached in the document, you can find also some minor suggestions.

Author Response

Q: 1) In the manuscript there are no information about semolina under study characteristics (protein amount, gluten quantity and quality). If it is possible at least to report the data shown in the product label this could be helpful, since protein amount is a very important information.

A: Added label information in section 2.1 raw material.

Q: 2) How did you choose the amount of water in every recipe? This is a very important parameter so an explanation of how you dosed it is required.

A: It is based on water contents in the juice and puree. Pasta samples should have the same hydration level for comparison. 28-33ml water per 100 gram solid content was chosen by other researchers, so 30ml of water per 100 gram solid content was choice for this study. As mentioned in line 53-56 in introduction. Achieve a high-level substitution of semolina by puree and juice need concentration, which does not suit our lab. The water for hydration come from vegetable juice or puree + addition water = 30 ml/100g. I have changed table 1 to make it clearer.

Q: 3) Regarding the references I think authors should consider also some other more recent studies about semolina dough ingredient characterization and substitution to compare with their formulation. Examples are:

Ahmad, N., Ur-Rehman, S., Shabbir, M. A., Shehzad, M. A., & Roberts, T. H. (2018). Fortification of durum wheat semolina with detoxified matri (Lathyrus sativus) flour to improve the nutritional properties of pasta. Journal of food science and technology55(6), 2114-2121.

De Pasquale, I., Verni, M., Verardo, V., Gómez-Caravaca, A. M., & Rizzello, C. G. (2021). Nutritional and functional advantages of the use of fermented black chickpea flour for semolina-pasta fortification. Foods, 10(1), 182.

Fanari, F., Desogus, F., Scano, E. A., Carboni, G., & Grosso, M. (2020). The effect of the relative amount of ingredients on the rheological properties of semolina doughs. Sustainability, 12(7), 2705.

Romano, A., Ferranti, P., Gallo, V., & Masi, P. (2021). New ingredients and alternatives to durum wheat semolina for a high quality dried pasta. Current Opinion in Food Science.

Schettino, R., Pontonio, E., & Rizzello, C. G. (2019). Use of fermented hemp, chickpea and milling by-products to improve the nutritional value of semolina pasta. Foods, 8(12), 604.

Xie, L., Nishijima, N., Oda, Y., Handa, A., Majumder, K., Xu, C., & Zhang, Y. (2020). Utilization of egg white solids to improve the texture and cooking quality of cooked and frozen pasta. LWT, 122, 109031.

A: Have added some in reference list.

Please check the revised draft for detail

Reviewer 2 Report

The authors present an interesting work about the effects of vegetable juice, puree and pomace on chemical and technological quality of fresh pasta. I appreciate this work; however, I request the authors to address the following comments.

Lines 13-14: Why do authors classify colour as a texture quality parameter?

Section 2.2: Why the storage temperatures for vegetable juice and puree were different (-28 and -18°C)?

Line 87: „uses” should be replaced with „using”.

Line 88: should be “red cabbage puree”

Line 94: „by mix” should be replaced with „by mixing”.

Line 95: „to extruded to pasta”?

Section 2.3: What was the key factor for choosing such a different vegetable compound concentrations (substitution level from 1 to 10)?

Section 2.4: There is no description of proximate analysis methods, but repetition of the pasta preparation procedure.

Section 2.6. Please provide the principle of the firmness test.

Lines 147-148: Do really L* parameter can have values from the range from -128 to +128?

Line 149-150: The sentence is not ended with dot.

Section 3.1 and Table 2: I recommend discussing first moisture (water) content, then protein content.

Table 2: In my opinion, fibre content should be provided. Authors discuss some changes based on fibre role in the pasta (e.g. line 306, 308-310), while those statement are not proven by the results of analysis. Moreover, providing a gluten content would be also valuable.

Table 2: Please provide water, protein, and ash contents for semolina.

Table 2: For comparison purposes, the data for uncooked and cooked pasta samples should be analysed statistically, in order to assess the effect of cooking on protein, starch, and ash contents in the samples.

Line 182: Should be “Protein, starch, and ash content (...).”

Line 169: Provide surnames of authors before [33].

Lines 173, 199, 219: “This/it may because” – please correct English.

Section 3.2: Start discussing the results from the OCT.

Lines 250-253: Rewrite the sentence.

Lines 253-254: “components of red cabbage pomace are less affinity to water” – please correct English.

Lines 270-271: “while other spinach pasta (...) distance” – correct English.

Line 273: “but not statistical significant”?, “SPO10 significantly”?

Lines 274-275: There are two separate sentences instead of one.

Line 276: Should be “a weakened structure”.

Line 277: Should be “compared”.

Line 276-278: Hydrocolloids such as locus bean gum or xanthan gum represent totally different characteristics than non-hydrocolloid components of spinach or red cabbage, so in my opinion, this reference is not proper.

Line 285: Should be “Figure 1 (c ) and figure 1 (d)”.

Line 290: Should be “was equal to”.

Figure 1 (captions): “The same letter is not significantly different from each other (p > 0.05)” – please correct (mean values are not significantly different).

Line 323: Should be “compared to”.

Lines 326-329: “After cooking, the red cabbage pasta trends to be even less bright, red, and yellow. Possibly due to the fact that the phytochemicals that provide the colour are water-soluble and leach into the cooking water” – leaching the coloured compounds to water would imply higher lightness – please verify.

Line 333: Should be “lead to acid evaporation”.

Line 342: Please provide more exhaustive title.

Table 4: Why the order of samples is different than in Tables 1-3?

Can authors propose the way of packaging of such a fresh vegetable pasta, that would preserve its texture and colour properties?

Author Response

Q: Lines 13-14: Why do authors classify colour as a texture quality parameter?

A: Move colour out from texture

Q: Section 2.2: Why the storage temperatures for vegetable juice and puree were different (-28 and -18°C)?

A: It is a typo. Thanks for checking it.

Q: Line 87: „uses” should be replaced with „using”.

A: It has changed to using

Q: Section 2.3: What was the key factor for choosing such a different vegetable compound concentrations (substitution level from 1 to 10)?

A: It is based on water contents in the juice and puree. Pasta samples should have the same hydration level for comparison. 28-33 ml water per 100 gram solid content was chosen by other researchers, so 30 ml of water per 100 gram solid content was choice for this study. As mentioned in line 53-56 in introduction. Achieve a high-level substitution of semolina by puree and juice need concentration, which does not suit our lab. 1% and 2% substitution is the level that can be both achieved by spinach and red cabbage material. 10% pomace is the sample act as a positive control as it expected to produce lowest techniqual quality. I have added some explanation in section 2.3

Q: Section 2.4: There is no description of proximate analysis methods, but repetition of the pasta preparation procedure.

A: Changed to proper text.

Q: Section 2.6. Please provide the principle of the firmness test.

A: test specific settings have been added.

Q: Lines 147-148: Do really L* parameter can have values from the range from -128 to +128?

A: changed to 100-0

Q: Line 149-150: The sentence is not ended with dot.

A: Added

Q: Section 3.1 and Table 2: I recommend discussing first moisture (water) content, then protein content.

A: as swelling index and water absorption index which related to moisture were discuss below, so moisture may not be discuss here.

Q: Table 2: In my opinion, fibre content should be provided. Authors discuss some changes based on fibre role in the pasta (e.g. line 306, 308-310), while those statement are not proven by the results of analysis. Moreover, providing a gluten content would be also valuable.

A: Fibre content have been tested, but it may be better to publish with glycaemic response data planned in the next article.

Q: Table 2: Please provide water, protein, and ash contents for semolina.

A: Data added to the table

Q: Line 182: Should be “Protein, starch, and ash content (...).

A: Changed

Q: Line 169: Provide surnames of authors before [33].

A: Added

Q: Lines 173, 199, 219: “This/it may because” – please correct English.

A: Corrected

Q: Lines 250-253: Rewrite the sentence.

Lines 253-254: “components of red cabbage pomace are less affinity to water” – please correct English.

A: Corrected to avoid misinterpretation

Q: Line 273: “but not statistical significant”?, “SPO10 significantly”?

A: Changed to make it clearer

Q:Lines 274-275: There are two separate sentences instead of one.

Line 276: Should be “a weakened structure”.

Line 277: Should be “compared”.

A: changed accordingly

Q: Line 276-278: Hydrocolloids such as locus bean gum or xanthan gum represent totally different characteristics than non-hydrocolloid components of spinach or red cabbage, so in my opinion, this reference is not proper.

A: Changed reference to non-hydrocolloids material

Q: Figure 1 (captions): “The same letter is not significantly different from each other (p > 0.05)” – please correct (mean values are not significantly different).

A: Figure caption changed to a clearer version

Q: Lines 326-329: “After cooking, the red cabbage pasta trends to be even less bright, red, and yellow. Possibly due to the fact that the phytochemicals that provide the colour are water-soluble and leach into the cooking water” – leaching the coloured compounds to water would imply higher lightness – please verify.

Line 342: Please provide more exhaustive title.

Table 4: Why the order of samples is different than in Tables 1-3?

A: lightness data cooked to uncooked, thank you very much for finding it out. Changed the table order and its title.

Q: Can authors propose the way of packaging of such a fresh vegetable pasta, that would preserve its texture and colour properties?

A: Sealed packing with a plastic bag stored at 4 °C would be cheaper to give around 7 day storage date. Cooked and then fast frozen, which is popular in some Asian countries like Japan, maybe one way to provide a convenient cooking experience would give a 1 year best before date stored at -18  °C.

Please check the revised draft for detail

Reviewer 3 Report

The manuscript reports an investigation of effect the substitution level of juice, puree and pomace, spinach and red cabbage on the pasta quality. It is an interesting study for gastronomy. The study design, methodologies used, and the interpretation of results are acceptable, but improvement can be made by addressing the following listed:

L10-11: This sentence is redundant. How much pasta with the addition of 1 g / 100g juce / pomace do you need to eat to provide vitamins, microelements, etc.?

L13-14: Colour is not a texture parameter

In the introduction, the authors state that drying reduces the nutritional value of vegetables added to pasta (L65-66). Then in the methodology (L83-84) they state that for 7 hours they dried the pomace and then ground it. The powder was added.

L98-100: is not clear

Table 1: Why after adding 19.37 g of vegetables, the dry matter from vegetables was 1 g?

from 8.73g was 2g?

Tabela 2. In the table, the units should be specified. g / 100g d.m. or g / 100g of material? It should be clear in the table. Use the same units for raw material and pasta.

If 100g of dry weight, for example, SJ, contains 38.56 protein and 21.82 ash? What are the other ingredients the dry weight of spinach and red cabbage? This is completely incomprehensible.

The nomenclature in Table 2 (a and b) should be corrected (Not Spinach pasta and spinach raw material nur spinach raw material and spinach pasta etc.)

L162-163: Raw material or paste is rich in protein?

L178: The legend is not correct. SJ, SPU, SPO are raw materials not pasta

L342: Change the order in table 4, first spinach then red cabbage.

L 353: „At a low substitution level, juice,  puree, and pomace can all be used ………” Please indicate what level is appropriate.

L357-358: Spinach and red cabbage were used, not other vegetables

Author Response

Q: L10-11: This sentence is redundant. How much pasta with the addition of 1 g / 100g juce / pomace do you need to eat to provide vitamins, microelements, etc.?

A: as shown in table 1 (revisioned), the addition is based on dry weight basis, such like SJ1-spinach juice 1%, the addition of vegetable is 21.49 g per 130 gram pasta. Commercial available vegetable pasta mostly used 1-3% of vegetable formula. The nutritional analysis (planed in following article) showed significant nutritional improvement when combined even 1% of vegetable material for cooked pasta. In vitro study is still undergoing.

Q: L13-14: Colour is not a texture parameter

A: moved colour out

Q: In the introduction, the authors state that drying reduces the nutritional value of vegetables added to pasta (L65-66). Then in the methodology (L83-84) they state that for 7 hours they dried the pomace and then ground it. The powder was added.

A: the pomace was used as a control to compare with puree and juice

Q: Table 1: Why after adding 19.37 g of vegetables, the dry matter from vegetables was 1 g?from 8.73g was 2g?

A: Thanks for finding out the problem, it is a mistake and have been corrected

Q: If 100g of dry weight, for example, SJ, contains 38.56 protein and 21.82 ash? What are the other ingredients the dry weight of spinach and red cabbage? This is completely incomprehensible.

A: table fixed, dry weight basis have been marked, fibre data was measured but planned to publish with glycaemic response data, so not present here.

Q: L178: The legend is not correct. SJ, SPU, SPO are raw materials not pasta

A: It have been corrected

Q: L342: Change the order in table 4, first spinach then red cabbage.

A: It have been corrected

Q: L 353: „At a low substitution level, juice, puree, and pomace can all be used ………” Please indicate what level is appropriate.

A: Specific level added

Please check the revised draft for detail

Round 2

Reviewer 3 Report

Is correct and clear.